# The role of π-blocking hydride ligands in a pressure-induced insulator-to-metal phase transition in $SrVO_2H$

Takafumi Yamamoto [1], Dihao Zeng[2], Takateru Kawakami[3], Vaida Arcisauskaite[2], Kanami Yata[3], Midori Amano Patino[2], Nana Izumo[1], John E. McGrady[2], Hiroshi Kageyama [1,4] & Michael A. Hayward[2]

Transition-metal oxyhydrides are of considerable current interest due to the unique features of the hydride anion, most notably the absence of valence $p$ orbitals. This feature distinguishes hydrides from all other anions, and gives rise to unprecedented properties in this new class of materials. Here we show via a high-pressure study of anion-ordered strontium vanadium oxyhydride $SrVO_2H$ that $H^-$ is extraordinarily compressible, and that pressure drives a transition from a Mott insulator to a metal at ~50 GPa. Density functional theory suggests that the band gap in the insulating state is reduced by pressure as a result of increased dispersion in the $ab$-plane due to enhanced $V_{d\pi}$-$O_{p\pi}$-$V_{d\pi}$ overlap. Remarkably, dispersion along $c$ is limited by the orthogonal $V_{d\pi}$-$H_{1s}$-$V_{d\pi}$ arrangement despite the greater $c$-axis compressibility, suggesting that the hydride anions act as π-blockers. The wider family of oxyhydrides may therefore give access to dimensionally reduced structures with novel electronic properties.

[1] Department of Energy and Hydrocarbon Chemistry, Graduate School of Engineering, Kyoto University, Nishikyo-ku, Kyoto 615-8510, Japan. [2] Department of Chemistry, University of Oxford, South Parks Road, Oxford OX1 3QR, UK. [3] Department of Physics, College of Humanities and Sciences, Nihon University, Setagaya, Tokyo 156-8550, Japan. [4] CREST, Japan Science and Technology Agency, 7-3-1 Hongo, Bunkyo-ku, Tokyo 113-0033, Japan. Takafumi Yamamoto and Dihao Zeng contributed equally to this work. Correspondence and requests for materials should be addressed to J.E.M. (email: john.mcgrady@chem.ox.ac.uk) or to H.K. (email: kage@scl.kyoto-u.ac.jp) or to M.A.H. (email: michael.hayward@chem.ox.ac.uk)

Transition-metal perovskite oxides, ABO$_3$, have been the subject of intense study, primarily due to their flexible chemistry which allows a wide variety of different metal cations to be incorporated within the same architecture. This fact, combined with the strong inter-cation electronic and magnetic coupling transmitted via O 2$p$ orbitals, makes the perovskite oxides ideal playgrounds to explore physical and chemical phenomena in transition-metal oxides[1–5]. The most common approach employed to tune the behavior and properties of perovskite oxides is chemical substitution, which typically involves cation substitutions on the A- or B-sites of the ABO$_3$ framework[6, 7]. However, modification of the anion lattice, either through the introduction of anion vacancies[8, 9], or by substituting non-oxide hetero-anions[10–12], provides further opportunities to make dramatic changes to their physical behavior. For example, aliovalent anion doping allows the adjustment of metal oxidation states, the modification of the on-site electronic configuration of transition-metal centers and the tuning of inter-cation couplings, all of which can lead to the realization of novel electronic states.

Binary metal hydrides (e.g., NaH and CaH$_2$) have proven particularly powerful reagents for modifying the anion lattice because they facilitate the de-intercalation of oxide anions. These low-temperature topochemical reactions[13, 14] give access to metastable, oxygen-deficient phases that contain transition-metal cations in extremely low oxidation states (e.g., Ni$^I$, Co$^I$, Mn$^I$, and Ru$^{II}$)[15–18] and/or highly unusual local coordination geometries (e.g., the square-planar Fe$^{II}$O$_4$ units observed in the widely studied infinite-layer phase, SrFeO$_2$)[19]. Reduction with metal hydrides can also bring about hydride-for-oxide anion exchange, yielding air-stable transition-metal oxyhydride phases, first observed during the conversion of LaSrCoO$_4$ and Sr$_3$Co$_2$O$_{7-x}$ into LaSrCoO$_3$H$_{0.7}$ and Sr$_3$Co$_2$O$_{4.33}$H$_{0.84}$, respectively[20, 21], and subsequently utilized to prepare ATiO$_{3-x}$H$_x$ (A = Ba, Sr, Ca)[22, 23] and Sr$_{n+1}$V$_n$O$_{2n+1}$H$_n$ ($n$ = 1, 2, ∞)[24]. Further transition-metal oxyhydride phases, such as SrCrO$_2$H and LaSrMnO$_{3.3}$H$_{0.7}$[25–27], have been prepared via high-pressure synthesis routes.

The incorporation of hydride anions into extended oxide phases can modify the properties of host phases dramatically due to the strongly contrasting features of oxide and hydride anions, the most obvious of which is the charge. For example, hydride-for-oxide substitution leads to reduction of the metal and induces metallic conductivity in insulating A$^{II}$TiO$_3$ phases[28, 29]. In addition, the lower electronegativity of hydride compared to oxide means a higher degree of covalency in M–H bonds compared to M–O, leading, for example, to strong Co–H–Co coupling via the σ framework in LaSrCoO$_3$H$_{0.7}$, and hence to a high magnetic ordering temperature[20, 30]. The labile nature of H$^-$ in mixed oxide/hydride lattices also allows for facile anion-exchange reactions with nitride sources (NH$_3$, for example), yielding mixed oxide-nitrides such as BaTiO$_{3-x}$N$_{2x/3}$[31, 32]. In the context of the present paper, however, the most significant difference between the hydride and oxide anions (or indeed any other anion) is the absence of filled valence orbitals with π-symmetry. As a result, when hydride is substituted for oxide in an extended solid, exchange pathways of π-symmetry are effectively blocked, changing the orbital connectivity of a phase quite dramatically, particularly when the anions are in an ordered arrangement. Hydride-for-oxide anion substitution, along with a resultant anion order, therefore offers the intriguing prospect of preparing phases with electronic configurations and properties that contrast strongly with the parent oxide phases.

Reaction of cubic SrVO$_3$ with CaH$_2$ yields tetragonal SrVO$_2$H, an oxide-hydride phase which adopts an anion-ordered, layered structure consisting of planes of vertex-linked V$^{III}$O$_4$ units separated by ordered SrH layers[24]. This hydride-for-oxide anion exchange converts the apex-linked V$^{IV}$O$_6$ units in SrVO$_3$ into trans-V$^{III}$O$_4$H$_2$ centers in SrVO$_2$H, lifting the degeneracy of the vanadium 3$d$ $t_{2g}$ orbitals[24]. On-site electron–electron repulsion further splits the half-filled $d_{xz/yz}$ orbitals, leading to an insulating ground state that has been observed in photoemission and X-ray absorption spectroscopy measurements on thin film samples[33]. Neutron diffraction measurements reveal G-type antiferromagnetic order at relatively high temperature ($T_N$ > 300 K), indicating the presence of strong antiferromagnetic interactions. The strong super-exchange coupling via the oxide ligands ($J_1$) is unremarkable, and has been confirmed by recent computational studies from Wei et al. and Liu et al. ($J_1$ = 42.7 meV and 23.5 meV, respectively)[34, 35]. In the absence of π-symmetry orbitals on the H$^-$ bridging ligands, inter-layer coupling ($J_2$) might be expected to be very small. The computational work by both Wei and Liu noted above confirms that the inter-layer coupling is indeed substantially less than its intra-layer counterpart, although it is not negligible ($J_2$ = 5.7 meV and 1.9 meV, respectively). SrVO$_2$H can therefore be viewed as a strongly correlated quasi-two-dimensional Mott insulator.

In this paper, we show how pressure can be used as a probe to investigate the structural and electronic properties of SrVO$_2$H. In the structurally related perovskite phase, SrFeO$_2$, an antiferromagnetically ordered Mott insulator at ambient pressure[19], the application of external pressure enhances the interactions between the FeO$_2$ layers, ultimately leading to a spin-state transition accompanied by antiferromagnetic-to-ferromagnetic and insulator-to-metal transitions[36]. Our motivation here is to establish how the presence of hydride ligands along the $c$-axis in SrVO$_2$H influences the pressure dependence of the structural and electronic properties of the perovskite lattice.

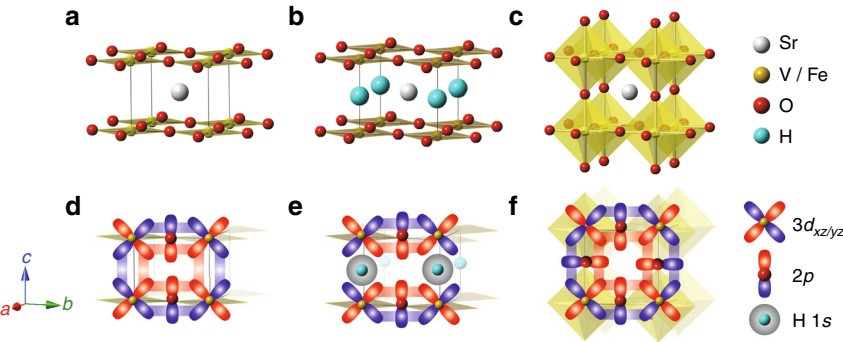

**Fig. 1** Crystal structures of perovskite related materials. **a** SrFeO$_2$, **b** SrVO$_2$H, and **c** SrVO$_3$. White, yellow, red, and sky blue spheres, respectively, denote Sr, transition metal, O and H atoms. **d–f** represent the inter- and intra-layer orbital connectivity

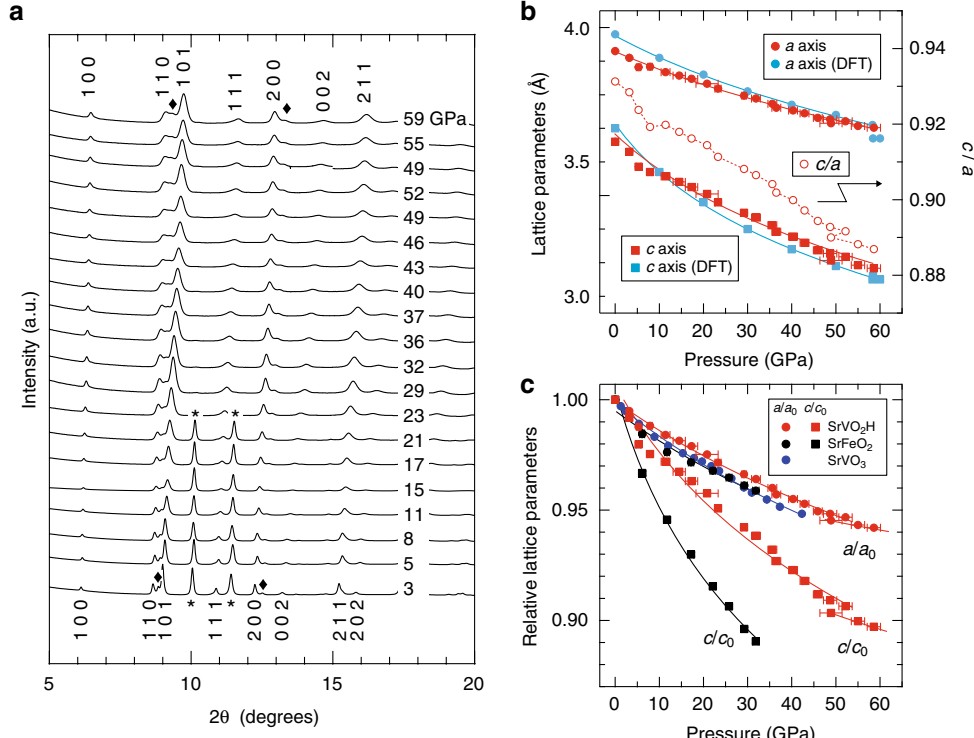

**Fig. 2** High-pressure behavior of SrVO$_2$H and SrFeO$_2$. **a** Powder synchrotron XRD patterns of SrVO$_2$H (sample SrVO$_2$H-A) under various pressures at room temperature. All the patterns are indexed on the basis of a tetragonal unit cell. The filled diamond and asterisk symbols correspond to SrVO$_3$ and rhenium from the gasket, respectively. **b** Pressure dependence of lattice parameters for the experimental (red) and the DFT-computed (sky blue) values of SrVO$_2$H – note that some error bars are smaller than the width of the symbols. The decrease in pressure from 52 GPa to 49 GPa as the cell volume decreases suggests a phase transition to a denser phase. **c** Relative lattice parameters, $a/a_0$ and $c/c_0$, of SrVO$_2$H (red), SrFeO$_2$ (black), and SrVO$_3$ (dark blue) as a function of pressure. Circles and squares correspond to the $a$ and $c$ axes, respectively. Solid lines in **b** and **c** represent linearized Birch-Murnaghan fits to the data

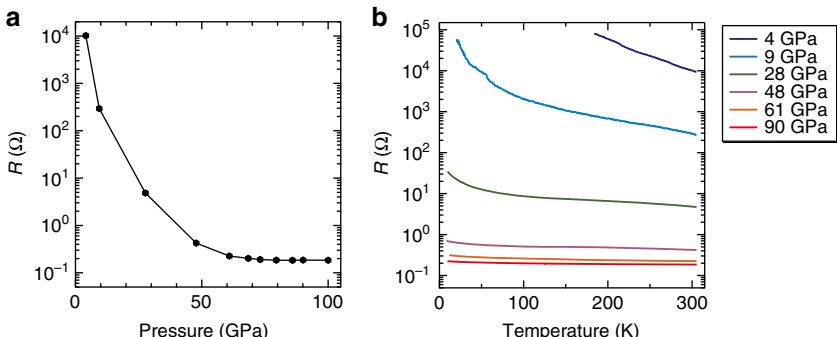

**Fig. 3** Electrical resistance for SrVO$_2$H under pressure. **a** Pressure dependence of resistance $R$ at 300 K (sample SrVO$_2$H-B). **b** Temperature dependence of resistance $R$ at various pressures

## Results

**Pressure-dependent structural studies of SrVO$_2$H.** SrVO$_2$H (Fig. 1b) was obtained from SrVO$_3$ (Fig. 1c) by using a CaH$_2$ reduction method. Powder synchrotron X-ray diffraction (XRD) data collected for SrVO$_2$H (sample SrVO$_2$H-A; Supplementary Note 1) at room temperature and a variety of applied pressures (Fig. 2a) can be readily indexed on the basis of a tetragonal unit cell, with no evidence of a symmetry-breaking structural transition up to the highest pressure measured (59 GPa). After releasing pressure, the pattern returned to the original ambient-pressure form with only slight peak broadening (Supplementary Fig. 2). The pressure dependence of the lattice parameters of SrVO$_2$H is shown in Fig. 2b and 2c, where the analogous data collected from

SrFeO$_2$[36] and SrVO$_3$ (Supplementary Fig. 3) are also shown for comparison. Linearized Birch-Murnaghan fits[37] to these data allow the zero-pressure linear compressibility $\beta$ of the $a$ and $c$ lattice parameters to be estimated (Supplementary Note 2). Fits to the data along $a$, which correspond to the compressibility of in-plane M–O bonds, yield similar $\beta_a$ values of $1.47(4) \times 10^{-3}$ Pa$^{-1}$ and $1.5(1) \times 10^{-3}$ GPa$^{-1}$, respectively, for SrVO$_2$H and SrFeO$_2$. These values are also similar to the isotropic compressibility of $1.72(4) \times 10^{-3}$ GPa$^{-1}$ for SrVO$_3$, indicating an approximately equal M–O bond stiffness in all three phases. In contrast, the $c$-axis compressibility of $\beta_c = 3.7(2) \times 10^{-3}$ GPa$^{-1}$ for SrVO$_2$H, while substantially lower than $\beta_c = 5.0(2) \times 10^{-3}$ GPa$^{-1}$ for SrFeO$_2$ where no ligand is present along the Fe–Fe vector

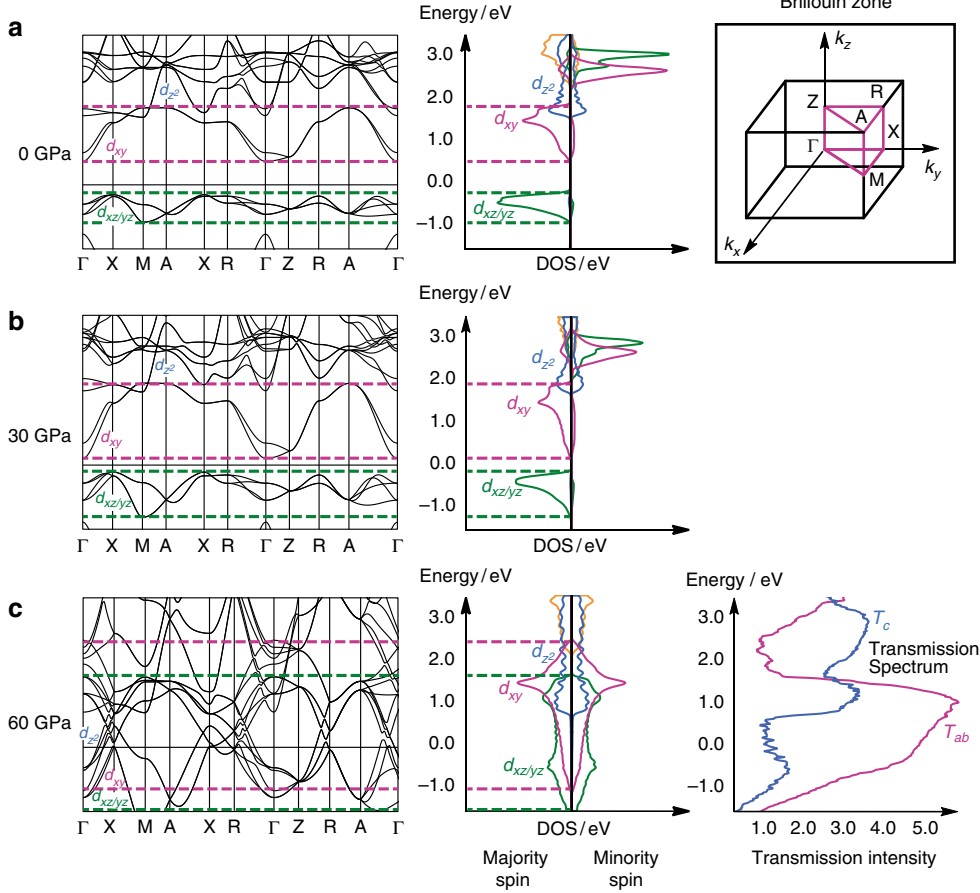

**Fig. 4** Density functional theory for SrVO$_2$H. DOS and band structure, **a** at ambient pressure, **b** at 30 GPa, and **c** at 60 GPa. The transmission spectrum along the c-axis ($T_c$, blue line) and along the V–O–V bisector ($T_{ab}$, purple) is also shown in plot **c** for the high-pressure state. Majority- and minority-spin states in the DOS are shown to the left and right of the axis, respectively. The first Brillouin zone for the tetragonal lattice is shown in the inset

(Fig. 1a), is far larger than any of the values of $\beta$ noted above, indicating that the V–H bonds in this phase are almost twice as compressible as the V–O bonds.

**A pressure-induced insulator-to-metal transition in SrVO$_2$H.** Figure 3 shows the resistivity of SrVO$_2$H (sample SrVO$_2$H-B; Supplementary Note 1) as a function of applied pressure. The observed resistivity of ~ 300 $\Omega$•cm at ambient pressure and 300 K is typical of a semiconductor and consistent with the Mott insulating picture proposed previously[24]. Application of up to 50 GPa pressure leads to a drop of four orders of magnitude in the resistivity of SrVO$_2$H at 300 K, and a change in the temperature dependence, $\Delta R/\Delta T$, such that the resistance becomes essentially independent of temperature at this pressure (Supplementary Fig. 4), indicating an insulator-to-metal transition. This behavior is reproducible across different samples (Supplementary Fig. 5). We note that $\Delta R/\Delta T$ does not become positive even at 90 GPa, as might be expected for a metallic phase. However, our measurements were performed on a non-sintered powder sample rather than a single crystal, so conduction across grain boundaries is likely to mask the metallic character. Similar features have been reported in previous studies[36, 38]. It should be noted that the absolute value of the resistivity is affected by the presence of impurities in the sample (such as SrVO$_{3-x}$), but the essential features of the insulator-to-metal transition can be identified in all cases (Supplementary Fig. 6). The precise value of the critical pressure, $P_c$, of the insulator-to-metal transition is not immediately obvious from the resistivity data, but a small but distinct

anomaly is observed in the plot of lattice parameters vs. pressure just below 50 GPa (Fig. 2b, 2c). The discontinuity in the plot arises because at this point a reduction in the volume of the sample space causes a decrease in the measured pressure, an observation that is consistent with a phase transition to a denser state (details in Supplementary Note 3). Our analysis of the electronic structure to be discussed in the next section suggests that this denser state may indeed be metallic.

**Ambient pressure electronic structure of SrVO$_2$H.** In order to explore the underlying origins of the physical measurements detailed above, we have turned to density functional theory. All calculations were performed at the PBE + $U$ level, with a $\sqrt{2} \times \sqrt{2} \times 2$ expansion of the unit cell as shown in Supplementary Fig. 8. We adopt a value of $U_{\text{eff}} = 2.0$ eV for the Hubbard $U$ parameter of vanadium, typical of similar calculations in the literature[39–41]. A full discussion of the influence of $U_{\text{eff}}$ on magnetic and structural properties, along with comparative data generated with the hybrid HSE06 functional, is presented in Supplementary Note 4, but our results are not strongly dependent on the chosen value, a point also noted by Wei et al. in their work[34]. The optimized cell parameters of the primitive unit cell at ambient pressure are $a = b = 3.98$ Å and $c = 3.69$ Å, in good agreement with the measured values of 3.93 Å and 3.67 Å. The spin densities on the vanadium centers confirm strong localization of the valence electrons, with $\rho(V) = \pm 1.73$ being typical of a formally $d^2$ ion in an oxide lattice. The density of states (DOS) at ambient pressure and its projection onto selected valence orbitals is shown

in Fig. 4a. The high spin moment on each metal center means that each of the metal $3d$ bands in the projected DOS (PDOS) is split into majority- and minority-spin components (shown to the left and right of the $y$-axes in Fig. 4, respectively). Of the vanadium-based $3d$ manifold, only the majority-spin $3d_{xz/yz}$ orbitals lie below the Fermi level, $E_f$, with their minority-spin counterparts being located ~ 3 eV above it. The strong splitting of the octahedral $t_{2g}$ manifold arises because whilst the $3d_{xz}$ and $3d_{yz}$ orbitals are destabilized by only two in-plane V–O $\pi^*$ interactions, $3d_{xy}$ is destabilized by four, and this anisotropy, combined with the finite $U_{eff}$, leads to the emergence of a band gap and hence to the semiconducting properties of SrVO$_2$H at ambient pressure. The band structure plot (Fig. 4a) highlights one further point that becomes significant in the context of the pressure-dependent properties: the dispersion of the bands just above and below $E_f$ ($d_{xz/yz}$ and $d_{xy}$) is more prominent in the $ab$-plane ($\Gamma \rightarrow X \rightarrow M$, $Z \rightarrow R \rightarrow A$) than along the $c$-axis ($M \rightarrow A$, $X \rightarrow R$, $\Gamma \rightarrow Z$). In direct contrast, the $d_{z^2}$ band is more strongly dispersed along $c$ due to σ-type interaction via hydride ligands, but the bottom of this band lies ~ 1.6 eV above $E_f$.

**Pressure dependence of the electronic structure.** The variation of optimized lattice parameters as a function of applied external pressure is displayed alongside the experimental data in Fig. 2b (sky blue lines). Whilst the computed lattice parameters are marginally more sensitive to pressure than those measured experimentally, the agreement between experiment and theory is striking, both in terms of the absolute values and in the differentiation between $a$ and $c$, where the compressibility along the latter is approximately double that along the former. The corresponding PDOS plots and band structure diagrams for selected pressures are summarized in Fig. 4b (30 GPa) and Fig. 4c (60 GPa). As the pressure increases to 30 GPa, the most conspicuous change in the DOS is that the vacant majority-spin $d_{xy}$ band broadens substantially (the dashed purple lines in Fig. 4 mark the upper and lower limits of this band), and the corresponding band structure confirms that this arises primarily due to increased dispersion in the $ab$-plane ($\Gamma \rightarrow X \rightarrow M$, $Z \rightarrow R \rightarrow A$). The occupied $d_{xz/yz}$ band also broadens, albeit to a lesser extent, and again this can be traced to greater dispersion in the $ab$-plane rather than along $c$ (compare changes in dispersion along $X \rightarrow M$ vs. $\Gamma \rightarrow Z$). At 60 GPa, the broadening is sufficient to induce overlap between the occupied $d_{xz/yz}$ and unoccupied $d_{xy}$ bands, resulting in the complete loss of spin density and the emergence of a metallic state with an associated discontinuity in the optimized lattice parameters, consistent with the features observed in the crystallographic and resistivity data around 50 GPa. Even at 60 GPa, it is clear that it is the dispersion in the $ab$-plane, and not along $c$, which is the driving force for the transition to a metallic state: a clear band gap is maintained along $\Gamma \rightarrow Z$ and $M \rightarrow A$. To reinforce this point, we have mimicked the effects of an anisotropic pressure source by varying either $a$ or $c$ while keeping the other constant (see Supplementary Fig. 11). If we maintain $a$ at its ambient pressure value of 3.93 Å but reduce $c$ to its 60 GPa-optimized value of 3.29 Å, we note that the dispersion of the $d_{xz/yz}$ band increases, but critically not to the extent that the band gap closes: the system remains an insulator. Conversely if we maintain $c$ at its ambient-pressure value of 3.69 Å but contract $a$ to its 60 GPa value of 3.67 Å, the band gap closes and a metallic state does emerge. From this, we conclude that it is the contraction in the $ab$-plane and not along $c$ that drives the formation of the metallic state, despite the fact that the magnitude of the contraction in $ab$ is smaller in absolute terms. To reinforce this point, we have explicitly computed the energy-dependent transmission spectrum

for current flow both along the $c$-axis and in the $ab$-plane for the 60 GPa structure (Fig. 4c, right). The major peak in $T_{ab}$ is coincident with the broad $d_{xy}$ band that spans the Fermi level, while transmission along $c$ ($T_c$) becomes significant only above ~ 0.8 eV, coincident with the bottom of the $d_{z^2}$ band. As a result, transmission at the Fermi level is approximately five times stronger in the $ab$-plane than along $c$, indicating that the metallic behavior is strongly two-dimensional. In summary, the emergence of metallic behavior appears not to be connected directly to the very substantial compression along $c$, which simply reflects the intrinsically higher compressibility of the hydride ion vs. oxide, rather than any increased delocalization of itinerant electrons along this axis. The high-pressure metallic state can instead be considered as a quasi-two-dimensional metal with a conduction band made up mainly of the overlapping $d_{xz/yz}$ and $d_{xy}$ bands with substantial dispersion only in the $ab$-plane. Finally we note that the precise window in which the insulator-to-metal transition is predicted to take place is strongly dependent on chosen methodology: a higher value of $U_{eff}$ pushes the transition to higher pressures, as does the hybrid HSE06 functional (see Supplementary Note 4 for a full discussion). We do not, therefore, claim to be able to pinpoint the critical pressure exactly, but rather note that the pressure-induced dispersion of the $d_{xy}$ band is the driving force in all cases. Thus unlike the absolute value of the critical pressure, our conclusions regarding the dimensionality of the metallic behavior are not strongly dependent on the chosen methodology.

## Discussion

At first sight the pressure-induced insulator-metal transition in SrVO$_2$H appears remarkably similar to that in SrFeO$_2$ previously reported by some of us[36] – both phases are antiferromagnetic insulators at ambient pressure, both undergo strongly anisotropic lattice contractions on the application of pressure ($\Delta c > \Delta a$) and both eventually become metallic, at 34 GPa and 50 GPa, respectively. A closer inspection however reveals profound differences between the two systems. In the case of SrFeO$_2$, the transition from insulating to metallic behavior is intimately connected to a change in spin state from $S = 2$ at ambient pressure to $S = 1$ above the critical pressure. The ambient-pressure electronic structure has been discussed at some length, and it is the $d_{z^2}$ orbital of Fe that is doubly occupied $((d_{z^2})^2 (d_{xz/yz})^2 (d_{xy})^1 (d_{x^2-y^2})^1)$[42, 43]. The band gap is primarily a consequence of the large spin polarization that splits the majority- and minority-spin manifolds. Above $P_c$ (34 GPa) however, the relative destabilization of the Fe–O $\sigma^*$ $3d_{x^2-y^2}$ orbital as the Fe–O bond lengths are compressed leads to a local $(d_{z^2})^2 (d_{xz/yz})^3 (d_{xy})^1 (d_{x^2-y^2})^0$ configuration ($S = 1$)[44]. At even higher pressures, the compression along $c$ leads to repulsions between the $d_{z^2}$ orbitals in adjacent layers, causing a partial transfer of electron density out of the $d_{z^2}$ band into $d_{xy}$, and therefore to a substantial increase in the DOS at $E_f$. The result is a metallic state where the electron density distribution is approximately isotropic[36]. In contrast, our calculations on the SrVO$_2$H system suggest that the greater compressibility along the $c$-axis is not directly related to the emergence of metallic behavior: even at the high-pressure limit the inter-layer V–V separation is too large to afford significant direct V$_{d\pi}$–V$_{d\pi}$ overlap. The very different behavior of SrFeO$_2$ and SrVO$_2$H arises simply because the electronic configuration is very different ($d^6$ vs. $d^2$) and the band gap in the latter is determined by the different number of V$_{d\pi}$–O$_{2p\pi}$–V$_{d\pi}$ interactions experienced by the $3d_{xy}$ and $3d_{xz/yz}$ orbitals (four and two, respectively). The hydride ions in SrVO$_2$H can therefore be viewed as π-orbital blockers in the sense that they have no π-symmetry valence orbitals to interact with the vanadium π-symmetry $d$-orbitals, $3d_{xz/yz}$. The metallic behavior

then emerges because the compression in the *ab*-plane, although smaller in magnitude than that along the *c*-axis, increases the $V_{d\pi}–O_{2p\pi}–V_{d\pi}$ interactions and hence the dispersion of the $3d_{xz/yz}$ and the $3d_{xy}$ bands, to the extent that they overlap and form a single continuous band. Schematic orbital connectivities for $SrFeO_2$, $SrVO_2H$ and $SrVO_3$ are shown in Fig. 2d–f. In short, when $SrVO_3$ is converted to $SrVO_2H$, the 3-dimensional $V_{d\pi}–O_{2p\pi}–V_{d\pi}$ network is reduced to two dimensions as a result of the π-blocking SrH layer, with dramatic consequences for the physical properties simply because the band gap falls in the middle of the $V_{d\pi}$ band.

Insulator-to-metal transitions in strongly correlated systems have been a central subject in condensed matter physics for many decades, and the transitions of a large number of oxides with different *d*-electron counts have been studied in some detail[45]. The physics of low *d*-electron count systems ($d^1$ or $d^2$ systems such as $VO_2$, $V_2O_3$, $Ti_2O_3$, and $REVO_3$ where RE = rare earth, for example) is, however, complicated because the splitting of the $M_{d\pi}$ manifold is very sensitive to the subtle details of the distortions and tilting of the $MO_6$ octahedra. In contrast, the highly anisotropic coordination environment of vanadium in $SrVO_2H$ simplifies the problem by causing a substantial zeroth-order splitting of this manifold, and the unusual pressure-dependent electronic properties of $SrVO_2H$ highlight the importance of the π-blocking nature of the hydride anion. We note that the use of π-blockers differs markedly from conventional strategies for reducing dimensionality, where bulky blocking layers are typically used to separate (super)conducting layers. These observations provide a strong motivation for the synthesis of new transition-metal oxyhydride phases with novel and elaborate anion-ordering schemes. Some closely related phases such as $Sr_2VO_3H$ and $Sr_3V_2O_5H_2$ (Supplementary Fig. 14) are already known[24, 27], and similar pressure-induced insulator-to-metal transitions might be anticipated in these S = 1 chains and ladders. Wider studies of the oxyhydride family will undoubtedly increase our ability to control π-orbital connectivity, in reduced phases and hence engineer materials with novel properties.

## Methods

**Sample preparation**. Powder samples of $SrVO_3$ were obtained by hydrogen reduction from $Sr_2V_2O_7$, as described previously[46]. Two powder samples of $SrVO_2H$ (referred to as $SrVO_2H$-A and $SrVO_2H$-B) were synthesized by a topochemical reaction between $SrVO_3$ and $CaH_2$, as reported previously[24]. The principal difference between the two reduced samples is that $SrVO_2H$-A contains small amounts of $SrVO_{3-x}$ which were observed to adversely affect the resistance measurements. A detailed discussion of samples $SrVO_2H$-A and $SrVO_2H$-B is presented in Supplementary Note 1.

**High-pressure X-ray diffraction**. Powder synchrotron XRD experiments for $SrVO_2H$-A under high pressures were performed at room temperature using the NE1A synchrotron beam line of the Photon Factory–Advanced Ring for Pulse X-rays (PF–AR) at the High Energy Accelerator Research Organization (KEK). Powder samples were loaded into a 100 μm hole of pre-indented rhenium gasket of the diamond anvil cell. Daphne oil 7373 was used as a pressure transmitting medium. The volume of the sample space was varied using a screw. At each volume, the fluorescence shift of ruby chips was used to measure the pressure. To estimate the pressure distribution along the sample, several ruby chips were placed inside the hole at different distances from its center. The pressure gradient increased with pressure, but did not exceed ±3 GPa at 59 GPa. The incident X-ray beam was monochromatized to a wavelength of 0.4173 Å. The beam was collimated to a diameter of about 50 μm at 0–21 GPa and 30 μm at 23–59 GPa. Due to the relatively large size of beam, peaks from the rhenium gasket were obvious in the pressure range 0–21 GPa. High-pressure experiments on $SrVO_3$ were conducted in the same manner, with a wavelength of 0.4186 Å and a beam collimated to 50 μm in diameter.

**High-pressure resistivity**. Four-probe ac resistance measurements were carried out using Pt electrodes. Three measurements were done on sample $SrVO_2H$-A up to 50 GPa and on sample $SrVO_2H$-B up to 100 GPa, all in the temperature range 8 K < T < 300 K. NaCl was used as a pressure transmitting medium. The sample-gasket cavity was coated with an insulating anhydrous gypsum ($CaSO_4$) with

epoxy. The initial sectional area and the distance between probes were about 40 × 40 $μm^2$ and 30 μm, respectively. The applied pressure was calibrated with fluorescence manometer on ruby chips placed around the sample.

**Density functional theory**. The geometry optimizations described here were performed using the VASP software package (VASP 5.3)[47], with the PBE density functional. In order to investigate different magnetic configurations it proved necessary to consider a $\sqrt{2} \times \sqrt{2} \times 2$ expansion of the primitive unit cell (Supplementary Fig. 8). A plane-wave cutoff of 600 eV was used and the Brillouin zone was sampled on an 8 × 8 × 7 Γ-centered grid. The effect of strong correlations is typically introduced through the effective Hubbard *U* value ($U_{eff}$). A survey of the literature indicates that $U_{eff}$ values up to 5.0 eV have been adopted for vanadium[39–41]. In the present study, we adopt a value of $U_{eff}$ = 2.0 eV, although the dependence of the optimized lattice parameters on $U_{eff}$ is weak (Supplementary Table 2). The potential pressure dependence of $U_{eff}$ in $SrFeO_2$ has been discussed, and Rahman et al. have suggested that its value should decrease approximately linearly with pressure from its ambient-pressure value[44]. Whilst systematic reduction of our chosen value of 2.0 eV to 0.0 eV in the present case reduces the absolute value of the critical pressure of insulator-to-metal transition, it does not influence the underlying physics in any substantial way. We have also considered the hybrid HSE06 functional[48] with 25% Hartree-Fock exchange and a range constant, ω, of 0.2. These calculations were done as single points using the geometries optimized at the PBE + U ($U_{eff}$ = 2.0 eV) level. An insulating state is retained with HSE06 even at a structure optimized under 90 GPa of pressure, but the closing of the band gap due to broadening of the $d_{xy}$ conduction band is clearly discernable. Transport analysis was performed using the ATK software package (ATK 2016.4)[49, 50] with the same PBE density functional and $U_{eff}$ = 2.0 eV used in the VASP calculations. A localized double-ζ basis set, extended with single polarization function (DZP) was used to describe all atoms. The Brillouin zone was sampled on a 21 × 21 × 21 Monkhorst-Pack grid.

**Data availability**. The data that support the findings of this study are available from the corresponding author upon reasonable request.

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

## Acknowledgements

The work was supported by CREST and JSPS KAKENHI (JP16H06438, JP16H6439, JP16H6440, JP16K21724, and JP16H06033). High-pressure synchrotron radiation experiments were performed under the approval of the Photon Factory Program Advisory Committee (No. 2015G012). We thank Taku Okada and Takumi Kikegawa for their support of the high-pressure XRD measurement. MAP thanks CONACYT and the Balliol College, Oxford for a scholarship. J.E.M. and V.A. acknowledge financial support from the EPSRC (EP/K021435/1).

## Author contributions

T.Y., D.Z., J.E.M., H.K., and M.A.H. designed the research. M.A.P. and N.I. synthesized the material. T.Y. collected high-pressure XRD data and analyzed. T.K. and K.Y. measured the high-pressure resistivity. D.Z. and V.A. performed the electronic structure analysis. All the authors discussed the results. T.Y., D.Z., H.K., J.E.M., and M.A.H. wrote the manuscript, with comments from all the authors.

## Additional information

**Competing interests:** The authors declare no competing financial interests.

