## [Peer Review File · Nature Communications]

Reviewers' Comments:

Reviewer #1:

Remarks to the Author:

Perovskite family has been one of the most technologically relevant groups of solid state compounds in which compositional engineering (doping or full substitution of ions) can be used to tune desirable, often unique physical properties and behavior. Crystal chemical robustness of the Pv structure opens a broad parameter space for such approach. Originally, combinatorial searches for unique compounds in Pv family have been mostly constrained to varying cations, but, as the authors of this manuscript point out, recent attempts at exchanging anions, to form e.g. oxynitrides, and oxyhydrides, in many cases lead to even more interesting results.

The manuscript in review deals with compressional behavior of one of such perovskite-type oxyhydride compounds, SrVO₂H, in which one of the oxygens is stoichiometrically replaced by a hydrogen. Throughout the manuscript authors make interesting comparisons between the series of related compounds: "normal" oxide perovskite SrVO₃, SrFeO₂ with four-coordinated iron, and the title oxyhydride compound, SrVO₂H. The paper describes a whole comprehensive suite of experiments and calculations, including synchrotron powder X-ray diffraction data, results of measurement of electrical resistance and first principles calculations exploring electronic band structure as a function of pressure.

At first sight the results do not seem surprising or overwhelmingly interesting, and I have to admit that on first reading I was a little puzzled as to why the authors chose to submit this work to Nature Communications. The compressibility up to the highest pressure reached, 60 GPa, is continuous and does not show obvious discontinuities. The strong elastic anisotropy (difference in compressibility between different crystallographic directions) is a logical consequence of the crystal structure and replacement of oxygen atoms with hydrogens along the (001) axis. The electrical resistance decreases very gradually, and seem to reach a point at which the sample becomes metallic around 50 GPa. All this is very typical for a normal robust semiconductor.

What makes this work interesting is the in-depth interpretation of the electronic origins of the insulator-metal transition enabled by the DFT results. Indeed it is quite intriguing to learn that metallic conductivity is enabled by the ab-plane interactions, and not by orbital overlap along the strongest compressing, c direction.

I have a few fairly minor technical comments.

I am surprised by the approach to estimate linear compressibility by doing a linear fit to unit cell parameters over a limited pressure range (this dependence is not linear). This has been used in literature in the past, but hardly seems satisfactory. More exact approach would be to use the results of linearized Birch-Murnaghan fit, and calculate axial compressibility as suggested in Reviews in Mineralogy and Geochemistry volume 41 (2000) by Hazen and Downs. The authors point out that DFT calculations predict noticeable discontinuity in unit cell parameter behavior at the insulator-metal transition. Indeed, this can be easily seen in Fig. 2 (green data point). On page 5 they also say that there is a "small but distinct anomaly" in the experimental data. I indeed see one anomalous point in Fig. 2 (orange data points), but at higher pressure the data seems to return to the previous trend. This is a little surprising, and makes me think that perhaps the experiment did not go to high-enough pressure to reach the transition (which could

have taken place at different pressure in diffraction and resistance experiments due to different nonhydrostaticity).

I do not understand why it is necessary to repeat the same figures (Fig. 3 and S4) twice in the main body of the paper, and supplementary materials, but with different pressure ranges? Perhaps these are experiments on samples A and B? Please, explain in the text or caption.

More technical details should be described for the electrical resistance measurement. I am assuming that no pressure medium was used, thus producing a much more anisotropic stress state than in diffraction experiments.

Figures 2b and 2c are not particularly useful together. It would be best to include 2c and add c/a ratio plot, which should be more sensitive to the discontinuity at the insulator-metal transition. I would also suggest including Birch-Murnaghan curves over whole range, instead of limited pressure-range linear fit curves for SrVO₂H in Figure 2.

Overall it is a very nicely written in-depth paper. I appreciate its value for enhancing understanding of oxyhydride modification approach to engineer electronic properties of perovskite materials. I think the paper will be suitable for publication in Nature Communications after revisions.

Reviewer #2:

Remarks to the Author:

The authors have studied samples of SrVO₂H synthesized by reacting SrVO₃ with CaH₂. They observe that under the application of high pressure the resistivity of these samples decreases and their insulating nature (as manifested by the dependence of resistivity on temperature) almost disappears. The structure of the samples determined by XRD remained tetragonal over the entire applied pressure range, with the c -axis lattice constant compressing much more than that of the a -axis. The structural data was used as a guide for calculating the band structures at different pressures that the authors then used to account for the observed insulator to metal transition. The transition was attributed to the changes in band structure due to lattice compression occurring along the a -axis; the c -axis compression by itself could not explain the transition based upon the band structure calculations.

The measurements and calculations presented in this manuscript appear to be well done and the manuscript itself is very well written. I am not a chemist like many of the authors but still found the work very intriguing and have been motivated by the manuscript to consider how hydride for oxide anion exchange could change the properties of the transition metal oxide systems that I have worked on in the past and am currently working with now, which I think says something about the broader audience that the paper would likely be of interest to.

I only have a few comments to make that hopefully are helpful:

1.) It would be nice to see the $R(T)$ curves for the data taken at the highest pressures in Fig. 3b on a linear resistance scale to confirm that they are indeed metallic. This would certainly strengthen the claim that an insulator to metal transition is present. In the current log-linear representation the $R(T)$ behavior is obscured but it appears that the resistance is still increasing slightly as temperature is reduced. If I am not wrong then how do the authors explain this? Their band structure calculations indicate band overlap (and a metallic state) should be achieved at the highest pressure.

2.) In Fig. 2, the last sentence of the caption specifies that triangles correspond to lattice parameters along M-[anion vacancies]-M but this data is not shown.

3.) A paper was published recently by T. Katayama *et al.* in the Journal of Applied Physics **120**, 085305 (2016) detailing measurements on thin films of SrVO₂H that the authors may wish to include in their references.

I recommend publication in Nature Communications provided the authors address the few comments above.

Reviewer #3:

Remarks to the Author:

The paper discusses the behavior of a recently synthesized oxyhydride compound, SrVO₂H, as a function of pressure. The material displays a pressure-induced insulator-metal transition initiated by the broadening of bands within the t_{2g} V-3d manifold that closes the small gap between out of plane d_{xz,yz} and in plane d_{xy} levels.

Electrical conductivity and structural determinations are coupled with DFT+U calculations.

As the authors correctly state in the final part of the discussion, strongly correlated materials have long been intriguing systems for advanced electronic structure studies, in both experiment and theory. The current material differs from others in that crystal field splitting and anion ordering effects reinforce each other in yielding effectively a system with 2D antiferromagnetic layers, isolated by the hydride ions that can't offer orbitals of pi symmetry to couple adjacent (001) layers.

Similar 2D behavior is observed in SrFeO₂ under pressure, although I found the comparison misleading, as structure, electronic configuration of the metal and details of the magnetic interaction all differ in the two materials and their effect cannot be deconvoluted. A more relevant comparison might have been found in SrCrO₂H, where the d³ configuration of Cr³⁺ opposed to the d² of V³⁺ yields different behavior, isotropic in the case of Cr that does not require Jahn-Teller like distortions.

There is certainly value in examining the SrVO₂H compound, but I don't think the system is of general enough interest to warrant publication of the manuscript in a Nature-family journal, and would like to refer the authors to a more specialized journal.

My concerns are linked:

1) to the instability of oxyhydride compounds, that are very unlikely to yield materials of practical interest for applications

2) the lack of analogue systems. The authors do mention further low-dimensional anion ordered oxyhydrides (in practice a Ruddlesden-Popper series) in the final sentence, but there is no indication on whether the materials would be (meta)stable and synthesizable, nor a prediction of which novel property they would display should synthesis succeed. Such a generalization and prediction would justify broader interest for the community.

A related concern is that in both Ruddlesden Popper phases quoted of Sr₂VO₃H and Sr₃V₂O₅H₂, V³⁺ would be in an average environment of 5 oxide and 1 hydride ligand, that does not necessarily yield stable anion ordering, which is necessary for the crystal field splitting of the V-3d levels at the basis of the behavior discussed here. We would therefore end up with a one-off situation.

In addition to the comments above, I would like to question the computational choice of DFT+U for

the electronic structure calculations. As the authors correctly mention, Mott insulators are critical cases where local DFT is inadequate, and require orbital dependent corrections. Results depend critically on the choice of the on-site parameter U . The more electronic states are localized, the higher the value of U to use, and since electronic localization is pressure dependent we should use a pressure-dependent value of U . I find this setting highly unsatisfactory.

The code used, VASP, allows for hybrid functionals such as HSE06, which apply an orbital-dependent correction to the electronic states determined self-consistently, and hence will respond to pressure-induced changes without modification of an input parameter.

In addition to submit the paper to a more specialized journal, I would also encourage the authors to include a hybrid functional study of the SrVO₂H band structure as a function of pressure.

Response to the reviewer's comments

Reviewer #1

Perovskite family has been one of the most technologically relevant groups of solid state compounds in which compositional engineering (doping or full substitution of ions) can be used to tune desirable, often unique physical properties and behavior. Crystal chemical robustness of the Pv structure opens a broad parameter space for such approach. Originally, combinatorial searches for unique compounds in Pv family have been mostly constrained to varying cations, but, as the authors of this manuscript point out, recent attempts at exchanging anions, to form e.g. oxynitrides, and oxyhydrides, in many cases lead to even more interesting results.

The manuscript in review deals with compressional behavior of one of such perovskite-type oxyhydride compounds, SrVO_2H , in which one of the oxygens is stoichiometrically replaced by a hydrogen. Throughout the manuscript authors make interesting comparisons between the series of related compounds: "normal" oxide perovskite SrVO_3 , SrFeO_2 with four-coordinated iron, and the title oxyhydride compound, SrVO_2H . The paper describes a whole comprehensive suite of experiments and calculations, including synchrotron powder X-ray diffraction data, results of measurement of electrical resistance and first principles calculations exploring electronic band structure as a function of pressure.

At first sight the results do not seem surprising or overwhelmingly interesting, and I have to admit that on first reading I was a little puzzled as to why the authors chose to submit this work to Nature Communications. The compressibility up to the highest pressure reached, 60 GPa, is continuous and does not show obvious discontinuities. The strong elastic anisotropy (difference in compressibility between different crystallographic directions) is a logical consequence of the crystal structure and replacement of oxygen atoms with hydrogens along the (001) axis. The electrical resistance decreases very gradually, and seem to reach a point at which the sample becomes metallic around 50 GPa. All this is very typical for a normal robust semiconductor.

What makes this work interesting is the in-depth interpretation of the electronic origins of the insulator-metal transition enabled by the DFT results. Indeed it is quite intriguing to learn that metallic conductivity is enabled by the ab -plane interactions, and not by orbital overlap along the strongest compressing, c direction.

Response: We would like to thank the referee for his/her summary of this study and the positive comments on the originality of our work. We are particularly encouraged that the referee fully appreciated the most significant issue relating to the anisotropy of the metallic character at high pressure. In order to reinforce this point, we have now explicitly computed the transmission along c and in the ab plane, and shown that at the Fermi level, the latter is greater by a factor of 5. The computed transmission spectra also map directly onto the density of states that we showed in the original manuscript, confirming that the in-plane transmission

is indeed dominated by the pressure-broadened $3d_{xy}$ band. One panel showing the transmission spectrum is added to Fig. 1c.

I have a few fairly minor technical comments. I am surprised by the approach to estimate linear compressibility by doing a linear fit to unit cell parameters over a limited pressure range (this dependence is not linear). This has been used in literature in the past, but hardly seems satisfactory. More exact approach would be to use the results of linearized Birch-Murnaghan fit, and calculate axial compressibility as suggested in Reviews in Mineralogy and Geochemistry volume 41 (2000) by Hazen and Downs.

Response: We thank reviewer #1 for this constructive suggestion. We agree that the linearized Birch-Murnaghan fit provides a more accurate method to calculate axial compressibility, and this approach is now used in Figure 2b and 2c. The axial moduli K_0 (inverse of compressibility) of 226 GPa and 90 GPa respectively for the a and c axes of SrVO₂H are different from the original results of linear fitting, but only subtly and the key message remains the same. The results confirm the key point that the c axis is twice as compressible as the a axis. To place these data in context, values of K_0 along the a axis of SrFeO₂ and SrVO₃ are 222 GPa and 194 GPa, respectively. In summary, the values obtained from the Birch-Murnaghan fit are reasonable and qualitatively unchanged from our previous linear fit analysis. The paragraph of “Pressure-dependent structural studies of SrVO₂H” (Pages 4 and 5) is modified accordingly, with the used formulas given in Supporting Information.

The authors point out that DFT calculations predict noticeable discontinuity in unit cell parameter behavior at the insulator-metal transition. Indeed, this can be easily seen in Fig. 2 (green data point). On page 5 they also say that there is a “small but distinct anomaly” in the experimental data. I indeed see one anomalous point in Fig. 2 (orange data points), but at higher pressure the data seems to return to the previous trend. This is a little surprising, and makes me think that perhaps the experiment did not go to high-enough pressure to reach the transition (which could have taken place at different pressure in diffraction and resistance experiments due to different nonhydrostaticity).

Response: We have considered this point carefully, and would like to take the opportunity to clarify exactly how these measurements were made. As shown in the left figure below, the volume of the cell is reduced by tightening a screw and the pressure in the cell is measured as an independent variable by observing the fluorescence shift of ruby chips as. In this case we used two different chips and both indicated a decrease of pressure at point 19 in the table below, despite a decrease in volume. Thus we believe that the decrease in pressure, and hence increase in density, at this point is real, and is caused by an IM (insulator-to-metal) transition. We do not believe that it is possible to establish whether the data return to the previous trend above this transition, as the referee suggests, because we have only 3 points above the transition. Thus we do not eliminate the possibility that the compressibilities of the metallic and insulating states are very different, but we do not have sufficient data at very high

pressures to prove it. However, we do note that very similar behaviour was observed in a structural transition in Sr_2PdO_3 (*Inorg. Chem.*, **50**, 11787–11794 (2011), as shown in the right figure. We have provided these experimental details in Supplementary Information with Table S1.

Table: Lattice parameters of SrVO_2H

No	Rotation of screw ($^\circ$)	Pressure (GPa)	a axis (\AA)	c axis (\AA)
1	—	0	3.93	3.66
2	0	3.2	3.90995	3.6302
3	30	5.3	3.88185	3.58585
4	45	7.9	3.88353	3.56996
5	60	11.4	3.86731	3.55725
6	75	14.5	3.85712	3.54061
7	105	17.3	3.84747	3.52509
8	120	20.8	3.83271	3.50477
9	135	23.3	3.8183	3.47979
10	165	29.2	3.79735	3.44848
11	180	31.9	3.78884	3.4346
12	195	35.6	3.77309	3.4114
13	210	36.5	3.76138	3.39232
14	225	40.3	3.7533	3.37758
15	240	42.9	3.74469	3.35957
16	255	45.9	3.73127	3.33716
17	270	48.7	3.72699	3.32747
18	285	52.2	3.72088	3.31771
19	300	48.9	3.71498	3.30642
20	315	55	3.70701	3.29268
21	330	58.6	3.70212	3.28343

I do not understand why it is necessary to repeat the same figures (Fig. 3 and S4) twice in the main body of the paper, and supplementary materials, but with

different pressure ranges? Perhaps these are experiments on samples A and B? Please, explain in the text or caption.

Response: Data shown in Fig. 3 and Fig. S4 of the original manuscript are obtained from independent runs for SrVO₂H-B. We conducted these experiments in order to check the reproducibility of the insulator-metal transition. This is explained in the main text as “Figure 3 shows the resistivity of SrVO₂H (sample SrVO₂H-B) as a function of applied pressure” and in Fig. S4 legend as “Note that this measurement reproduces the results shown in Fig. 3, but represents an independent run” For further clarification, we have added a sentence, “The measurement was performed on SrVO₂H-B” in Fig. 3 legend. Different pressure ranges are due to economical reason. Since an experiment over 50 GPa has a high risk of breaking diamond anvil, we stopped one of the experiments at 50 GPa.

More technical details should be described for the electrical resistance measurement. I am assuming that no pressure medium was used, thus producing a much more anisotropic stress state than in diffraction experiments.

Response: We used NaCl as a pressure transmitting medium for the electrical resistance measurement. Since we did not specify this point in the original manuscript, we have modified corresponding sentences in Experimental Methods.

Figures 2b and 2c are not particularly useful together. It would be best to include 2c and add c/a ratio plot, which should be more sensitive to the discontinuity at the insulator-metal transition. I would also suggest including Birch-Murnaghan curves over whole range, instead of limited pressure-range linear fit curves for SrVO₂H in Figure 2.

Response: We have modified Fig. 2 along the lines suggested by the referee: In Fig. 2b, the c/a ratio was added. For clarity, theoretical data on SrVO₂H are shown only in Fig. 2b, while experimental data on SrFeO₂ and SrVO₃ are shown only in Fig. 2c. In both Fig. 2b and 2c, linearized Birch-Murnaghan fits below P_c are shown. One reference is added (Ref. 37).

Overall it is a very nicely written in-depth paper. I appreciate its value for enhancing understanding of oxyhydride modification approach to engineer electronic properties of perovskite materials. I think the paper will be suitable for publication in Nature Communications after revisions.

Reviewer #2

The authors have studied samples of SrVO₂H synthesized by reacting SrVO₃ with CaH₂. They observe that under the application of high pressure the resistivity of these samples decreases and their insulating nature (as manifested by the dependence of resistivity on temperature) almost disappears. The structure of the

samples determined by XRD remained tetragonal over the entire applied pressure range, with the c-axis lattice constant compressing much more than that of the a-axis. The structural data was used as a guide for calculating the band structures at different pressures that the authors then used to account for the observed insulator to metal transition. The transition was attributed to the changes in band structure due to lattice compression occurring along the a-axis; the c-axis compression by itself could not explain the transition based upon the band structure calculations.

The measurements and calculations presented in this manuscript appear to be well done and the manuscript itself is very well written. I am not a chemist like many of the authors but still found the work very intriguing and have been motivated by the manuscript to consider how hydride for oxide anion exchange could change the properties of the transition metal oxide systems that I have worked on in the past and am currently working with now, which I think says something about the broader audience that the paper would likely be of interest to.

Response: We would like to thank the referee for the positive endorsement and for recognizing the value of our study.

I only have a few comments to make that hopefully are helpful:

It would be nice to see the $R(T)$ curves for the data taken at the highest pressures in Fig. 3b on a linear resistance scale to confirm that they are indeed metallic. This would certainly strengthen the claim that an insulator to metal transition is present. In the current log-linear representation the $R(T)$ behavior is obscured but it appears that the resistance is still increasing slightly as temperature is reduced. If I am not wrong then how do the authors explain this? Their band structure calculations indicate band overlap (and a metallic state) should be achieved at the highest pressure.

Response: The reviewer is correct to point out that $\Delta R/\Delta T$ does not become positive even at the highest pressure (90 GPa), although the resistance does become essentially independent of temperature above ~ 50 GPa, indicating that an insulator-metal transition occurs. Our DFT calculations indicate that the conductivity is extremely anisotropic, and so it will be extremely difficult to observe a positive temperature dependence for a non-sintered powder specimen such as the one used here. We have noted similar characteristics in previous publications (*Nat. Chem.* **1**, 371–376 (2009) and *J. Am. Chem. Soc.* **133**, 6036–6043 (2011), the latter added as Ref. 38). We have amended Figure S4 and included sentences in the text to clarify this issue (page 5 line 13-16).

2.) In Fig. 2, the last sentence of the caption specifies that triangles correspond to lattice parameters along M -[anion vacancies]- M but this data is not shown.

Response: The Figure legend was modified.

3.) A paper was published recently by T. Katayama et al. in the Journal of Applied Physics **120**, 085305 (2016) detailing measurements on thin films of SrVO₂H that the authors may wish to include in their references.

Response: We added a sentence in the introduction (page 3 line 16-17) with the suggested paper cited as Ref. 33.

I recommend publication in Nature Communications provided the authors address the few comments above

Reviewer #3

The paper discusses the behavior of a recently synthesized oxyhydride compound, SrVO₂H, as a function of pressure. The material displays a pressure-induced insulator-metal transition initiated by the broadening of bands within the t_{2g} V-3d manifold that closes the small gap between out of plane d_{xz,yz} and in plane d_{xy} levels.

Electrical conductivity and structural determinations are coupled with DFT+U calculations.

As the authors correctly state in the final part of the discussion, strongly correlated materials have long been intriguing systems for advanced electronic structure studies, in both experiment and theory. The current material differs from others in that crystal field splitting and anion ordering effects reinforce each other in yielding effectively a system with 2D antiferromagnetic layers, isolated by the hydride ions that can't offer orbitals of pi symmetry to couple adjacent (001) layers.

Similar 2D behavior is observed in SrFeO₂ under pressure, although I found the comparison misleading, as structure, electronic configuration of the metal and details of the magnetic interaction all differ in the two materials and their effect cannot be deconvoluted. A more relevant comparison might have been found in SrCrO₂H, where the d³ configuration of Cr³⁺ opposed to the d² of V³⁺ yields different behavior, isotropic in the case of Cr that does not require Jahn-Teller like distortions.

Response: We think that the reviewer has not fully appreciated the role of SrFeO₂ as a comparison phase for the behavior of SrVO₂H under pressure. Our contention in this manuscript is that the combination of a stoichiometric (1:1) exchange of hydride for oxide and an ordered oxide-hydride arrangement in SrVO₂H, allows us to observe highly unusual physical behavior in this phase. Namely, a pressure-induced insulator-to-metal transition which is driven by the small compression of the *ab*-plane of the material rather than the much larger compression observed in the *c*-axis, with the resulting metallic phase having a strong

2-dimensional character.

We have chosen SrFeO₂ as a comparator for the behavior of SrVO₂H despite the very obvious differences in electronic configuration at the metal specifically because this iron phase is crystallographically ordered (ordered anion vacancies) so the ‘local’ electronic state of the FeO₄ centres is well defined. On the application of pressure SrFeO₂ becomes metallic due to a *c*-axis compression, thus it could be considered the ‘normal’ case. In SrFeO₂, compression along the *c* axis involves the close approach of two ‘lone pairs’ – the pairs of electrons in the *s/d_{z²}* hybrids. In SrVO₂H, in contrast, the V centres are separated by a hydride ligand (in effect a single pair of electrons combined with one proton). The comparison of the compressibilities was therefore intended to highlight the fundamentally different electronic environments along the *c* direction.

The reviewer suggests a comparison to SrCrO₂H would be more informative than to SrFeO₂. However, SrCrO₂H has a disordered O/H lattice as well as a d³ rather than d² electron count, so again it could be argued that it is not possible to deconvolute these effects. Furthermore, while SrCrO₂H has a isotropic cubic structure on average, locally each Cr centre has an O₄H₂ coordination so the three ‘t_{2g}’ orbitals will not be degenerate and the system will not be isotropic on the local scale and this disorder could well dominate the behavior of the phase. Again we stress the key point that we are seeking to illustrate that unlike oxide ions or anion vacancies, hydride ions have the ability to ‘block’ the π-orbital connectivity of a system. Oxide-hydrides should therefore be considered as qualitatively different to ‘vacant’ oxides or other mixed oxy-anion systems.

There is certainly value in examining the SrVO₂H compound, but I don't think the system is of general enough interest to warrant publication of the manuscript in a Nature-family journal, and would like to refer the authors to a more specialized journal.

My concerns are linked:

1) to the instability of oxyhydride compounds, that are very unlikely to yield materials of practical interest for applications

In response, we would like to emphasize the fact that there are many solid systems which are metastable, or at least unstable in air, but have still yielded important physical insights. The superconducting iron oxy-arsenide and oxy-selenide phases, the lead-halide perovskites and graphene spring immediately to mind. If we restrict study to compounds which are ‘stable’ by conventional measures, we miss a great deal of interesting chemistry and physics.

Moreover, the oxide hydride phases are more stable than might appear at first sight. While it is indeed likely that most transition-metal oxide-hydride phases are not thermodynamically stable at ambient pressure, many of the known oxide-hydride phases are kinetically stable at room temperature and resist reaction with oxygen and water for long periods of time. We simply added “air-stable” to the introductory part (page 2, 19).

2) the lack of analogue systems. The authors do mention further low-dimensional anion ordered oxyhydrides (in practice a Ruddlesden-Popper series) in the final sentence, but there is no indication on whether the materials would be (meta)stable and synthesizable, nor a prediction of which novel property they

would display should synthesis succeed. Such a generalization and prediction would justify broader interest for the community.

A related concern is that in both Ruddlesden Popper phases quoted of Sr₂VO₃H and Sr₃V₂O₅H₂, V³⁺ would be in an average environment of 5 oxide and 1 hydride ligand, that does not necessarily yield stable anion ordering, which is necessary for the crystal field splitting of the V-3d levels at the basis of the behavior discussed here. We would therefore end up with a one-off situation.

The Ruddlesden-Popper phases, Sr₂VO₃H and Sr₃V₂O₅H₂ that the referee refers to are in fact known phases (*Angew. Chem. Int. Ed.* **53**, 7556–7559 (2014)), prepared in the same way as SrVO₂H. Both contain VO₄H₂ units, but they are connected into lower dimensional arrangements than SrVO₂H. In general, mixed oxide-hydride phases are becoming increasingly common in the literature, and systems containing Ti, V, Cr, Mn and Co have already been reported in the literature. We believe that it is a matter of time before other transition elements are added to this list, and so we do not share the reviewer's pessimism about the limited scope of these materials. On the contrary, we believe that the proven existence of many air-stable oxyhydrides and analogous systems with trans-V³⁺O₄H₂ octahedra proves that the oxyhydrides system is of general enough interest to warrant publication of the manuscript in *Nature Communications*. To avoid a potential confusion, the crystal structures of Sr₂VO₃H and Sr₃V₂O₅H₂ are provided in Figure S14. The final paragraph has also been substantially modified, as well as to emphasise the importance of the control of dimensionality. The abstract has been also updated so that it explains the context and general interest in what we hope is a more transparent manner.

In addition to the comments above, I would like to question the computational choice of DFT+U for the electronic structure calculations. As the authors correctly mention, Mott insulators are critical cases where local DFT is inadequate, and require orbital dependent corrections. Results depend critically on the choice of the on-site parameter U. The more electronic states are localized, the higher the value of U to use, and since electronic localization is pressure dependent we should use a pressure-dependent value of U. I find this setting highly unsatisfactory.

The code used, VASP, allows for hybrid functionals such as HSE06, which apply an orbital-dependent correction to the electronic states determined self-consistently, and hence will respond to pressure-induced changes without modification of an input parameter. In addition to submit the paper to a more specialized journal, I would also encourage the authors to include a hybrid functional study of the SrVO₂H band structure as a function of pressure,

We absolutely agree with the referee that the choice of U is critical. Our original choice, $U = 2.0$ eV, is consistent with other work in the literature for low-oxidation state early transition metals, but there is undoubtedly a degree of arbitrariness here. In the original manuscript we noted that the DFT results placed the metal-insulator transition in the 30-60 GPa window, mapping on to an experimental value of ~ 50 GPa. This may have given the incorrect impression that we believe the DFT numbers in an absolute sense. Motivated by the referee's comment, we have undertaken a more systematic study of the effects of U and, as the referee suggests above, also with the hybrid HSE06 functional. We should note that the latter are extremely demanding on hardware resources, and we have therefore done HSE06

calculations only as single points on geometries optimised using our standard PBE+ U model. It is also fair to note that the HSE06 functional, as with all hybrids, carries a degree of parameterisation in the choice of the exact exchange (25%) and also the range parameter ($\omega = 0.2$), so in some sense we are trading an arbitrary choice of U for an arbitrary choice of exact exchange. The hybrid functional certainly favours the insulating state to a greater degree than our original $U = 2.0$ eV calculations, to the extent that even at 90 GPa a band gap is retained. Nevertheless, the gap is systematically reduced as pressure increases, and the narrowing of the gap can be traced to the broadening of the d_{xy} band, precisely the points we highlighted in the original discussion. We conclude, therefore, that the choice of functional does indeed control the precise pressure at which the metal-to-insulator transition occurs, but it does not alter the qualitative changes in band structure that drive the transition. Whilst these calculations with HSE06 data are certainly informative, we do not feel that they fundamentally change the key message of this paper, so they are included in the supporting information with two new figures (Figures S12 and S13). We deleted the second paragraph of the section “Ambient pressure electronic structure of SrVO₂H” as it is now less important.

Reviewers' Comments:

Reviewer #1:

Remarks to the Author:

I read the revised manuscript carefully and am very happy with the modifications. The minor technical issues indicated in the first round of reviews have been adequately resolved and the clarity of the presentation has been noticeably improved. I can now recommend the paper for publication in Nature Communications.

Reviewer #2:

Remarks to the Author:

The authors have addressed my comments and questions in a reasonable manner and I think the manuscript should be accepted.

Reviewer #3:

Remarks to the Author:

The authors have addressed the main concerns raised by the three reviewers.

Since my concern related to the suitability of the paper to Nature Communication was not shared by the other two reviewers, I am happy for the paper to be published in the journal.

I would still like to clarify one issue with the authors, perhaps to clarify my own understanding of data presented.

As I suspected, computed band structures and band gaps are heavily dependent on the DFT functional chosen; indeed trends are qualitatively reproduced, but the pressure of any insulator-metal transition has extremely large variation depending on the method chosen for the study.

Given the variation of computed data, the main structural hint we have for the insulator-metal transition rests in the XRD data of figures 2 and conductivity measurements of S15-6. While the decrease of lattice parameters at ~50GPa appears in figure 2b (corresponding to a decrease of pressure (table S1) between a screw rotation of 285 to 300 degrees), this point is not present in the XRD patterns. Can the authors please clarify this issue, and provide the additional XRD pattern corresponding to the discontinuity? Ideally XRD patterns in figure 2a should be labelled with the screw rotation as for data in the newly provided table S1, so that a correspondence of data for figure 2a-c and table S1 can be obtained unambiguously. A similar

discontinuity in pressure as a function of screw rotation occurs between 195-210 degrees, but this is not discussed. As a computational chemist I am out of my comfort zone here, but I'd like to be reassured by the authors that all experimental accidents can be ruled out at both pressure vs angle discontinuities.

Figure 2 data refer to the sample labeled A; have the corresponding measurements been made for sample B as well?

The comparison would be of interest, since the measured conductivity of samples A and B varies by over 2 orders of magnitude (compare figures S5 and S6) and has distinctly different T dependence, which may hint to a large sample dependence of results that would be visible in XRD patterns too.

While qualitative results are discussed in my view correctly, and are of interest, I still find that an improvement in the quantitative discussion is needed before the paper can be accepted in a Nature quality publication.

I'm happy for final acceptance of the paper to be an editorial decision, once the authors address the comments above.

Response to the reviewer's comments

Reviewer #3

The authors have addressed the main concerns raised by the three reviewers.

Since my concern related to the suitability of the paper to Nature Communication was not shared by the other two reviewers, I am happy for the paper to be published in the journal.

I would still like to clarify one issue with the authors, perhaps to clarify my own understanding of data presented.

As I suspected, computed band structures and band gaps are heavily dependent on the DFT functional chosen; indeed trends are qualitatively reproduced, but the pressure of any insulator-metal transition has extremely large variation depending on the method chosen for the study.

Given the variation of computed data, the main structural hint we have for the insulator-metal transition rests in the XRD data of figures 2 and conductivity measurements of SI5-6.

While the decrease of lattice parameters at ~50GPa appears in figure 2b (corresponding to a decrease of pressure (table S1) between a screw rotation of 285 to 300 degrees), this point is not present in the XRD patterns. Can the authors please clarify this issue, and provide the additional XRD pattern corresponding to the discontinuity? Ideally XRD patterns in figure 2a should be labelled with the screw rotation as for data in the newly provided table S1, so that a correspondence of data for figure 2a-c and table S1 can be obtained unambiguously.

Response: To address these questions, we need to clarify the precise details of the pressure-dependent crystallography experiments. These are done by rotating a screw which controls the distance between the diamond anvils (i.e. volume of the sample space). The pressure and the X-ray diffraction pattern are then measured independently at each point. When a sample volume (lattice parameter) changes in a discontinuous manner, a discontinuity in the pressure can occur. Thus, the drop of pressure we found suggests a phase transition to denser phase. We modified sentences in the section "A pressure-induced insulator-to-metal transition in SrVO₂H." in page 5 of the main text and Supplementary Note 3.

As the reviewer pointed out that, a correspondence of data for figure 2a-c and table S1 was unambiguous in the former version. We have added the additional XRD pattern corresponding to the discontinuity in Fig. 2 and 2S. Also, Run No. (corresponds to Run No. in Table 1S) is added in Figure 2S. So, Fig. 2, Fig2S, and table S1 are now fully consistent.

A similar discontinuity in pressure as a function of screw rotation occurs between 195-210 degrees, but this is not discussed. As a computational chemist I am out of my comfort zone here, but I'd like to be reassured by the authors that all experimental accidents can be ruled out at both pressure vs angle discontinuities.

Response: The reviewer is correct to note that the pressure at a rotation angle of 210 does indeed lie somewhat below the line connecting the two neighbouring points. However the pressure does not actually decrease at this point, as it clearly does at 300, and we believe that this apparent discontinuity reflects the error in the measurement rather than a phase change. We show the pressure-vs-rotation-curve with a red linear fitting curve between 5 GPa to 52 GPa below (the same figure was attached in Supplementary Figure 7).

Figure 2 data refer to the sample labeled A; have the corresponding measurements been made for sample B as well?

The comparison would be of interest, since the measured conductivity of samples A and B varies by over 2 orders of magnitude (compare figures S5 and S6) and has distinctly different T dependence, which may hint to a large sample dependence of results that would be visible in XRD patterns too.

Response: We did not perform the high pressure XRD measurement for the sample B. The high pressure XRD measurements for the sample A were performed only at room temperature, so the corresponding measurements for sample B do not give information for the different T dependence of the conductivity data. We note that, as shown in the XRD patterns shown below, the sample B is nearly pure SrVO_2H while the sample A has ~14wt% of SrVO_3 impurity. Since metallic SrVO_3 has significant influence on the resistance, the sample A shows much lower resistance. The different T dependence also derived from different impurity amount.

While qualitative results are discussed in my view correctly, and are of interest, I still find that an improvement in the quantitative discussion is needed before the paper can be accepted in a Nature quality publication.

Response: We hope that the above discussion clarifies the major sources of confusion for reviewer 3, and that the clearer description of the experimental protocol will avoid similar confusion for readers. It is not entirely clear to us what the reviewer is looking for here. We have evidence for an insulator to metallic transition from the resistance data, and evidence to support a phase transition of some kind from the discontinuity in the XRD data. Even without the discontinuity, it is clear that the compressibility is highly anisotropic, consistent with the reduced dimensionality of the crystal. The DFT calculations tie the various threads of evidence together – the pressure dependence of the computed lattice parameters reproduces the experimental observations, both in terms of the anisotropy of the compressibility and the potential for a transition to a metallic state. Most importantly, it emphasise the point that the compression in the *ab* plane is most important, despite the greater compressibility along *c*. As the reviewer has pointed out, DFT is unable to give a quantitative statement on exactly where a phase transition will take place – the critical pressure for the insulator-metal transition is highly dependent on methodology, and the only way we can benchmark it is against the discontinuity in the XRD data. We hope that we have been open about this point in the manuscript, and have not made unduly optimistic claims for predictive accuracy.